# Adaptive laboratory evolution reveals regulators involved in repressing biofilm development as key players in *Bacillus subtilis* root colonization

Maude Pomerleau,[1] Vincent Charron-Lamoureux,[1] Lucille Léonard,[2] Frédéric Grenier,[1] Sébastien Rodrigue,[1] Pascale B. Beauregard[1]

**ABSTRACT** Root-associated microorganisms play an important role in plant health, such as plant growth-promoting rhizobacteria (PGPR) from the *Bacillus* and *Pseudomonas* genera. Although bacterial consortia including these two genera would represent a promising avenue to efficient biofertilizer formulation, we observed that *Bacillus subtilis* root colonization is decreased by the presence of *Pseudomonas fluorescens* and *Pseudomonas protegens*. To determine if *B. subtilis* can adapt to the inhibitory effect of *Pseudomonas* on roots, we conducted adaptive laboratory evolution experiments with *B. subtilis* in mono-association or co-cultured with *P. fluorescens* on tomato plant roots. Evolved isolates with various colony morphology and stronger colonization capacity of both tomato plant and *Arabidopsis thaliana* roots emerged rapidly from the two evolution experiments. Certain evolved isolates also had better fitness on the root in the presence of other *Pseudomonas* species. In all independent lineages, whole-genome resequencing revealed non-synonymous mutations in genes *ywcC* or *sinR* encoding regulators involved in repressing biofilm development, suggesting their involvement in enhanced root colonization. These findings provide insights into the molecular mechanisms underlying *B. subtilis* adaptation to root colonization and highlight the potential of directed evolution to enhance the beneficial traits of PGPR.

**IMPORTANCE** In this study, we aimed to enhance the abilities of the plant-beneficial bacterium *Bacillus subtilis* to colonize plant roots in the presence of competing *Pseudomonas* bacteria. To achieve this, we conducted adaptive laboratory experiments, allowing *Bacillus* to evolve in a defined environment. We successfully obtained strains of *Bacillus* that were more effective at colonizing plant roots than the ancestor strain. To identify the genetic changes driving this improvement, we sequenced the genomes of these evolved strains. Interestingly, mutations that facilitated the formation of robust biofilms on roots were predominant. Many of these evolved *Bacillus* isolates also displayed the remarkable ability to outcompete *Pseudomonas* species. Our research sheds light on the mutational paths selected in *Bacillus subtilis* to thrive in root environments and offers exciting prospects for improving beneficial traits in plant growth-promoting microorganisms. Ultimately, this could pave the way for the development of more effective biofertilizers and sustainable agricultural practices.

**KEYWORDS** *Bacillus subtilis*, *Pseudomonas fluorescens*, root colonization, adaptive laboratory evolution, biofilm regulation, YwcC, SinR

N umerous microorganisms live and thrive close to plant roots and form a distinct microbiota playing an important role in promoting plant health (1). Certain plants were shown to actively select beneficial bacteria in the rhizosphere via the secretion of specific compounds in their exudates (2), stimulating an interaction that benefits both

**Ad Hoc Peer Reviewers** Ákos T. Kovács, Universiteit Leiden, Leiden, Netherlands; Yunrong Chai, Northeastern University, Boston, Massachusetts, USA

Address correspondence to Pascale B. Beauregard, pascale.b.beauregard@usherbrooke.ca.

The authors declare no conflict of interest.

See the funding table on p. 14.

the plants and bacteria. One of these microorganisms is *Bacillus subtilis,* a Gram-positive, soil-ubiquitous bacteria defined more than 40 years ago as a plant growth-promoting rhizobacterium (PGPR) (3). Many strains from the *Bacillus subtilis* species complex are widely used in agricultural settings due to their multiple beneficial effects on plants (4–7). They can have direct effects, such as increasing nutrient bioavailability and stimulating plant growth via the production of phytohormones (8, 9). They can also serve as biocontrol agents, since secondary metabolites such as plipastatin and surfactin have antimicrobial activity against pathogens (7, 10–12). *B. subtilis* can also prime the plant defense system by triggering the induced systemic resistance (2).

While many PGPR strains show strong, reproducible plant enhancements in controlled conditions, biofertilizer effects are variable in field conditions (13–16). Different factors can explain this lack of efficacy, such as the extent of the inoculant biological activity in the receiving matrices (17, 18). For example, *Bacillus subtilis* was shown to rapidly sporulate when cells were introduced in bulk soil, thereby limiting their beneficial effect. Certain plants, either grown sterile or with other microorganisms, can produce exudates, which modulate *B. subtilis* sporulation (19, 20). A second factor is the inoculant capacity to establish in the rhizosphere in the presence of the residing microbiota (17, 18). Multiple interactions with the soil community can result in the inhibition of important traits for *B. subtilis* rhizosphere establishment (21, 22).

One key factor for *B. subtilis* root colonization is its capacity to form biofilms on roots (6, 23, 24). The biofilm extracellular matrix of *B. subtilis* is composed of exopolysaccharides (EPS), protein fibers (TasA and TapA), and a small hydrophobin protein (BslA) (4, 25). Poly-y-glutamic acids, a secreted polymer, were also reported to play an important role in biofilm robustness and root colonization (26). In non-biofilm conditions, operons encoding for EPS synthesis and TasA/TapA are repressed by the transcriptional repressor SinR (4, 25). Plant polysaccharides and malic acids, which are respectively present at the surface of the root and in root exudates, trigger a signaling pathway resulting in SinR inhibition and expression of matrix genes (23, 27).

Adaptive laboratory evolution (ALE) experiments are a powerful tool for studying microbial adaptation to specific environments (28). Performed in mono-association with a plant, these assays can lead to the emergence of strong mutualist traits and reveal the molecular mechanisms involved in colonization (29–32). The laboratory evolution of *B. subtilis* on *Arabidopsis thaliana* led to the emergence of evolved isolates of various morphologies with increased root colonization by themselves or in combination (30, 31). Some mutations were found to influence biofilm formation in the presence of xylan and decreased motility, suggesting an evolutionary trade-off between those traits (30, 32). One isolate was also shown to increase root colonization within a synthetic community (30).

Many *Pseudomonas* species are also characterized as PGPR, and several were co-isolated with *Bacillus* from the rhizosphere where they likely interact with each other (33–35). Even if the combination of these two genera can increase biocontrol abilities and plant growth, most interactions between them were reported to be antagonistic *in vitro* (22, 36–39). For example, certain *Pseudomonas* can repress or disperse *B. subtilis* biofilm via the production of the secondary metabolite 2,4-diacetylphloroglucinol and rhamnolipids (21, 40, 41). These interactions likely impact *B. subtilis* capacity to strongly establish on plant roots.

In this study, we first observed that *Pseudomonas* species impaired the establishment of *B. subtilis* on the roots of *A. thaliana* and tomato seedlings. To determine if *B. subtilis* can evolve to overcome this challenge and what traits would be involved, we elaborated an ALE on tomato roots in which *B. subtilis* was either challenged with *Pseudomonas* or not. Evolved isolates with enhanced root colonization rapidly emerged. In both evolution experiments, multiple mutations were identified in *ywcC* and *sinR,* genes encoding the regulators involved in the repression of biofilm development, which suggests that the ability to compete with *Pseudomonas* for root colonization is highly linked to biofilm formation.

## RESULTS

### *Pseudomonas* spp. impede *B. subtilis* colonization on seedlings

*In vitro* antagonism between *Pseudomonas* species and *B. subtilis* was often reported, but no studies have yet shown how these interactions translated on plant roots (21, 22, 36, 42). We thus performed root colonization assays with *B. subtilis* NCIB 3610 in the presence of three *Pseudomonas* species that are used as biocontrol agents (*Pseudomonas fluorescens* WCS365, *Pseudomonas capeferrum* WCS358, and *Pseudomonas protegens* Pf-5) (43, 44). *Arabidopsis thaliana* and tomato, grown in hydroponic conditions, were used for this assay since the outcome might be plant-specific. *B. subtilis* colonization was quantified by colony-forming unit (CFU) enumeration for *A. thaliana*, while the dense biofilm formed on tomato roots made this technique imprecise. Consequently, we used an alternative quantitative method where *B. subtilis* constitutively expressed the fluorescence gene *mKate2,* and the relative fluorescence unit present on the root was measured (see Fig. S2). As shown in Fig. 1, *P. protegens* and, to a lesser extent, *P. fluorescens* decreased *B. subtilis* presence in both plants, while *P. capeferrum* influenced *B. subtilis* establishment on *A. thaliana* only. These results point toward a general, plant-independent inhibitory effect of *Pseudomonas* on *B. subtilis* root colonization. This observation is further corroborated by a recent *in vitro* biofilm study showing that pseudomonads are very likely to inhibit *B. subtilis* pellicle in a low-nutrient medium and, in particular, species producing 2,4-diactylphloroglucinol such as *P. protegens* (45).

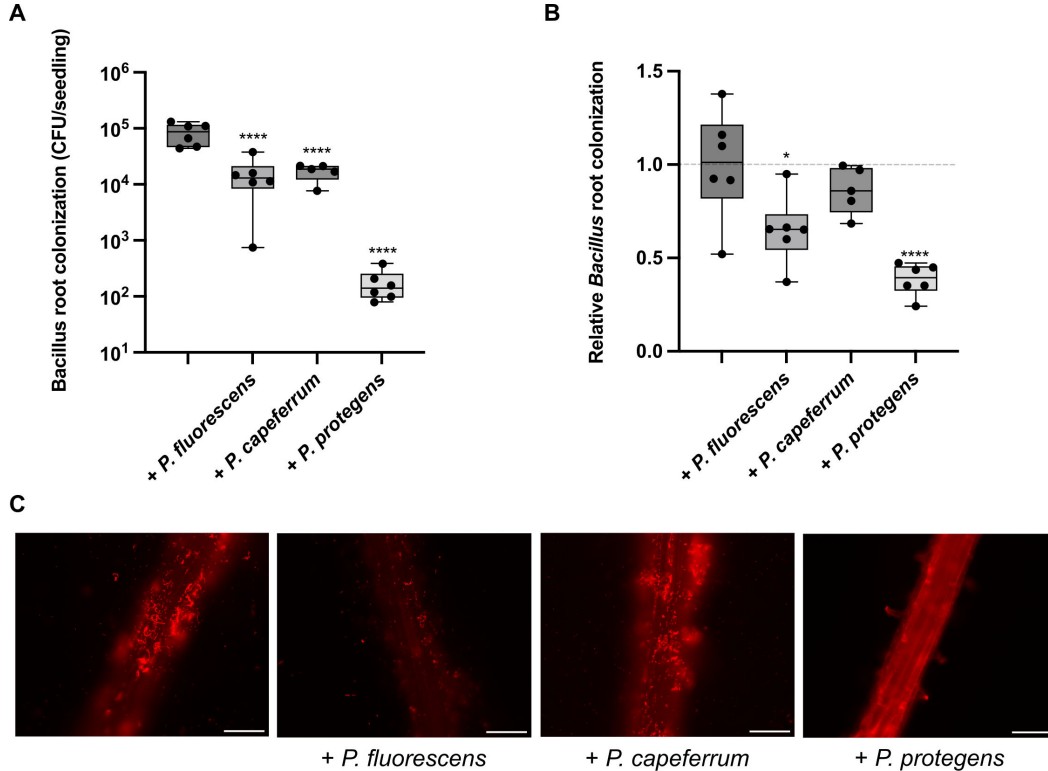

**FIG 1** *B. subtilis* colonization is decreased by *Pseudomonas* species. (A) Quantification of *B. subtilis* colonization on 6-day-old *A. thaliana* roots when inoculated alone or with different *Pseudomonas*. (B) Relative quantification of tomato root colonization by *B. subtilis* NCIB 3610 constitutively expressing mKATE2 and inoculated with *Pseudomonas* spp., reported as a ratio over the mean fluorescence intensity of *B. subtilis* colonizing the plant alone $^{*}P < 0.05$; $^{****}P < 0.0001$, ANOVA followed by Dunnett's *post hoc* test. (C) Representative pictures of *A. thaliana* root colonization when *B. subtilis* is inoculated alone or with different *Pseudomonas*. The scale bar is 100 µm for all images.

## *Bacillus subtilis* evolves rapidly on tomato plant roots

To determine if *B. subtilis* could adapt to root colonization in the presence of a competitor, we performed a directed evolution assay in which we selected for *B. subtilis* cells able to colonize the root. Briefly, the ancestral strain *B. subtilis* NCIB 3610 was inoculated on tomato roots before being actively dispersed by sonication (30% for 30 s), diluted, and transferred to a new root every 24 h for a total of 21 cycles (Fig. 2A). The experiment was performed with *B. subtilis* alone (*Bacillus*-root evolution [BRE]) or with the addition of *Pseudomonas fluorescens* WCS365 acting as a competitor (*Bacillus*-root-*Pseudomonas* evolution [BRPE]). For BRPE, *P. fluorescens* WCS365 was added 24 h after *B. subtilis* was inoculated and incubated for an additional 24 h before the cells were detached by sonication (40% for 60 s, which kills *P. fluorescens*), diluted, and transferred to a new root (Fig. 2A). This strategy was chosen to help determine if the acquired mutations were specific for improved colonization or competition or shared between both evolution assays. *P. fluorescens* was chosen as the competitor strain in the BRPE experiment due to its moderate impact on *B. subtilis* colonization. We followed the colonization of the six lineages (six independent plants) at each cycle by evaluating the colonization of *B. subtilis* on the root (Fig. 2B and C). Of note, one of the lineages with *P. fluorescens* was lost due to contamination and is consequently not shown. Interestingly, the BRPE showed an upward trend in *B. subtilis* colonization capacity, while the BRE had a drop in the first cycles and ended approximately at the same colonization capacity as the starting point (Fig. 2B and C).

As previously observed by Blake et al. (31), we noticed that different *B. subtilis* morphological variants emerged during the evolution and rapidly dominated the population with the ancestral strain morphology disappearing only a few cycles after the first appearance of a new variant. After 7, 14, and 21 strain cycles, six evolved isolates per lineage were cryopreserved by selecting differently shaped colonies after plating. The morphologies of colonies seeded from cultures from the frozen stocks were then compared, and we selected at least one strain of each morphology present in a lineage at a specific cycle to be sequenced. A total of 78 evolved isolates (36 for evolution alone and 42 for paired evolution) were sequenced. The analysis revealed that 31 of these isolates were clones of at least one other isolate within lineages, and 12 isolates were a perfect match of the ancestral strain. Of note, the ancestral strain was also re-sequenced to establish a strong basis for the analysis. Single-nucleotide polymorphisms (SNPs) were the most frequent type of mutation for both evolution experiments, representing 69.4% of all mutations, compared with 3.2%, 3.2%, and 24.2% for insertions, synonymous variants, and deletions, respectively. Overall, 62 different mutations were identified, of which 39 were found in the BRPE, 19 with the BRE, and 3 shared between both (Tables S1 and S2). In addition to the presence of the competitor, the harshest sonication in the BRPE compared to the BRE might have also impacted the acquisition of mutations. Even though more mutations were acquired by isolates evolved with *P. fluorescens*, the ratios of SNPs over other different types of mutations between both evolution experiments were very similar (70% vs 69%). Of note, the population size of the BRPE was smaller than the population size of BRE for certain cycles, but not consistently.

Our analysis revealed that the BRPE led to more isolates with two or more mutations (17 out of 27) than the BRE (6 out of 13). Interestingly, all lineages from BRE bared a conserved non-synonymous mutation that appeared before the seventh cycle, contrasting with the paired evolution, where only two lineages maintained a conserved mutation from cycles 7 to 21 (Tables S1 and S2). All those observations suggest that mutations take more cycles to be fixed in a tri-partite system, with the root and *Pseudomonas* (BRPE) than with the root alone (BRE), possibly due to a more complex selection pressure.

## Mutations in biofilm regulation genes are conserved among evolved isolates

By cycle 21, we observed that non-synonymous mutations in the negative biofilm regulator genes *sinR* or *ywcC* were found in all lineages. Interestingly, while evolved strains with mutations in *ywcC* displayed similar "rough" morphotypes, this was not the

**A**

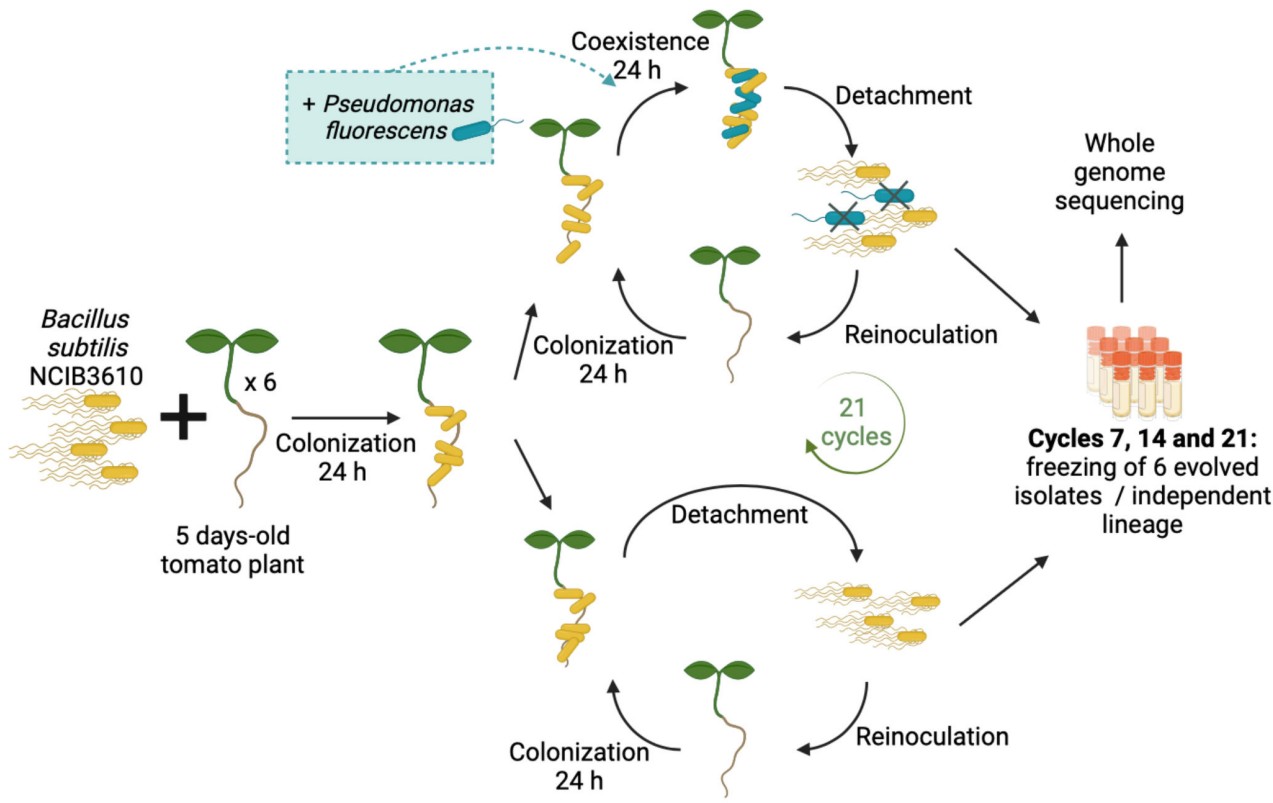

**B** **C**

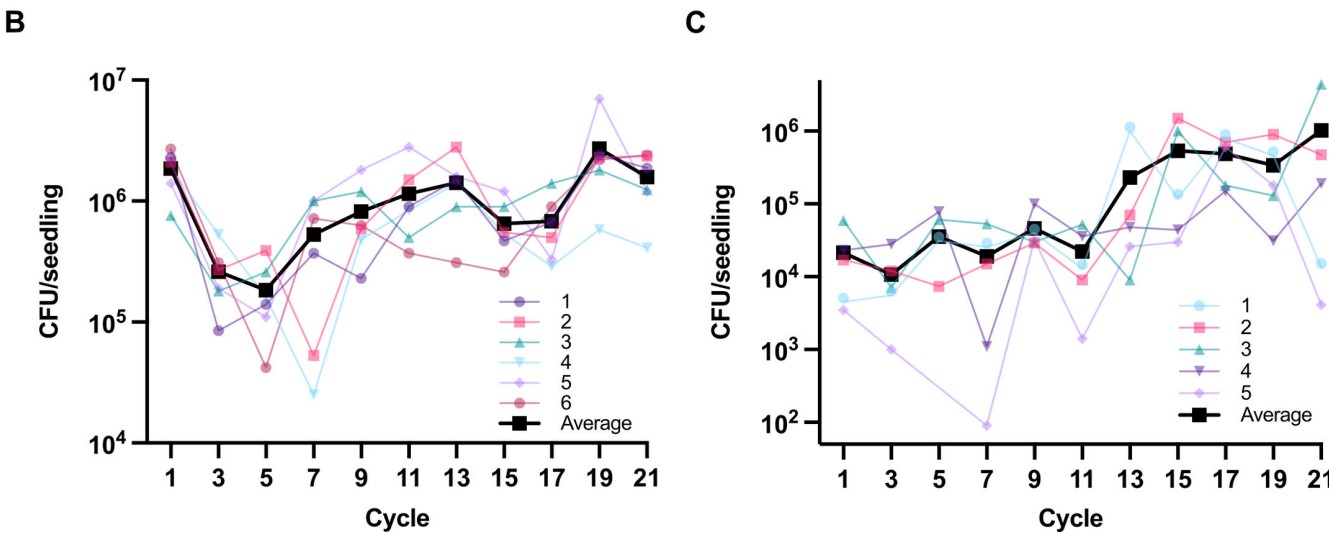

**FIG 2** Evolution of *B. subtilis* on tomato roots. (A) Depiction of the directed evolution of *B. subtilis* in the presence of *P. fluorescens* WCS365 or alone on tomato plant roots. (See Materials and Methods for a detailed description of the experiment.) (B) CFU of *B. subtilis* colonizing the root every two cycles for the experimental evolution alone or (C) with *P. fluorescens*. (A) was created with BioRender.com

case for *sinR* mutants. For the BRE, conserved non-synonymous mutations in *ywcC* appeared in three independent lineages, and non-synonymous mutations in *sinR* appeared in the three other lineages (Table S1). In the BRPE, non-synonymous mutations in the two genes were also found in three lineages each but were not mutually exclusive

since one lineage contained non-synonymous mutations in both genes (Table 2; Table S2). The apparition of these non-synonymous mutations in parallel evolutions and

TABLE 1  Non-synonymous mutations identified in the genomes of isolates evolved alone (BRE) on tomato roots

| Evolution alone | | | | | | |
|---|---|---|---|---|---|---|
| Evolved isolates | | | | | Gene | Mutation |
| A1a.C7 | A2.a.C14 | A3.a.C7 | A4.d.C7 | A6.a.C7 | | |
| X | | | | | *ywcC* | 214_215delAA  Frameshift variant |
| | | X | | | | 250G>T  Stop gained |
| | | | X | | | 11delA  Frameshift variant |
| | X | | | | *sinR* | 125C>T  Missense variant |
| | | X | | | | 296T>C  Missense variant |
| | | X | | | *mccA* | 632G>T  Missense variant |

lineages suggests that they confer an advantage to root colonization. We chose to further examine one evolved isolate per lineage for most lineages (BRE; Table 1) and three pairs of evolved isolates, each coming from a different lineage, which displayed increased non-synonymous mutations throughout the cycles (BRPE; Table 2). The latter allowed us to examine the effect of a single non-synonymous mutation versus a combination.

## Evolved isolates show enhanced colonization in a plant-independent manner

The competitivity of the evolved isolates was first tested on tomato plants with or without *P. fluorescens,* in an experimental setup similar to the directed evolution assay. As shown in Fig. 3A, most evolved isolates showed a twofold increase in root colonization capacity when inoculated alone. Similar results were obtained for colonization assays in the presence of *P. fluorescens*, even from isolates that were evolved without the selective pressure of the competitor (Fig. 3B). This observation suggests that the capacity to compete with *P. fluorescens* strongly correlates with an enhanced root biofilm formation capacity.

Hu et al. (32) recently observed that different mutations in *B. subtilis* evolved in the presence of various plant species, and some of them were detected more frequently when the evolution was performed on *A. thaliana* versus tomato plant roots. Another recent study observed that isolates evolved on *A. thaliana* roots did not have an improvement in the root colonization of tomato plants (31). Here, as shown in Fig. 4, our tomato-evolved isolates from both evolutions appeared to have some increased capacity to colonize *A. thaliana* roots. The results are, however, more variable, which might be due to the strength of the phenotype. Curiously, only two evolved isolates showed significant improvement of *A. thaliana* root colonization when inoculated alone (Fig. 4A), but many showed increased colonization compared to the ancestral strain when *P. fluorescens* was present (Fig. 4B). These observations suggest that the evolution on tomato roots does

TABLE 2  Non-synonymous mutations identified in the genomes of isolates evolved with *P. fluorescens* (BRPE) on tomato roots

| Evolution with *P. fluorescens* | | | | | | | |
|---|---|---|---|---|---|---|---|
| Evolved isolates | | | | | | Gene | Mutation |
| P3.a.C7 | P3.a.C14 | P4.a.C7 | P4.b.C14 | P5.a.C14 | P5.d.C21 | | |
| X | X | X | | | | *ywcC* | 11delA  Frameshift variant |
| | X | | | | | *sinR* | 296T>C  Missense variant |
| | | | X | X | | | 170C>T  Missense variant |
| | | | | | X | *dxs* | 1246G>T  Missense variant |
| | X | | | | | *dxr* | 386C>T  Missense variant |
| | | X | X | | | *pksR* | 5041T>C  Missense variant |

**A**

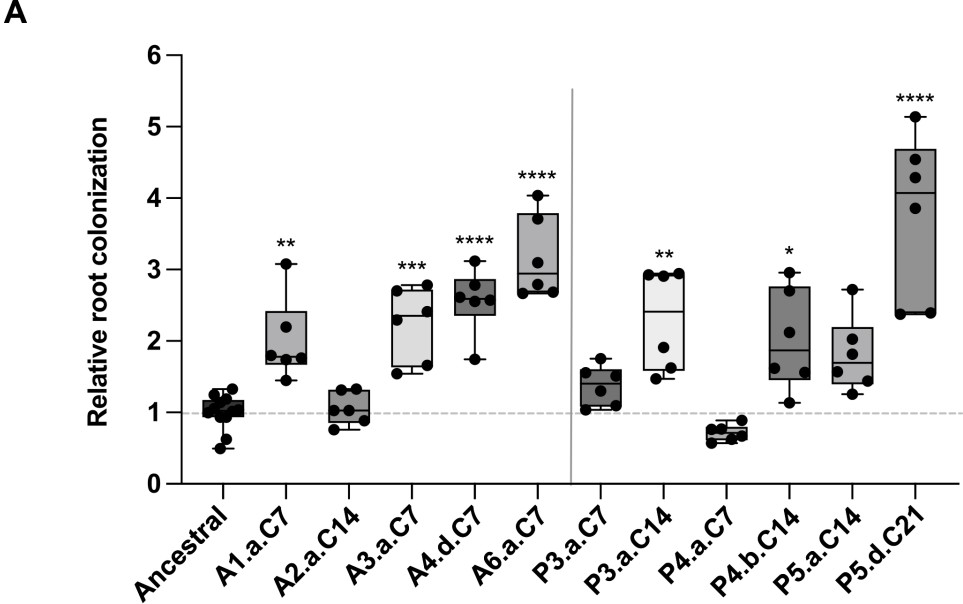

**B**

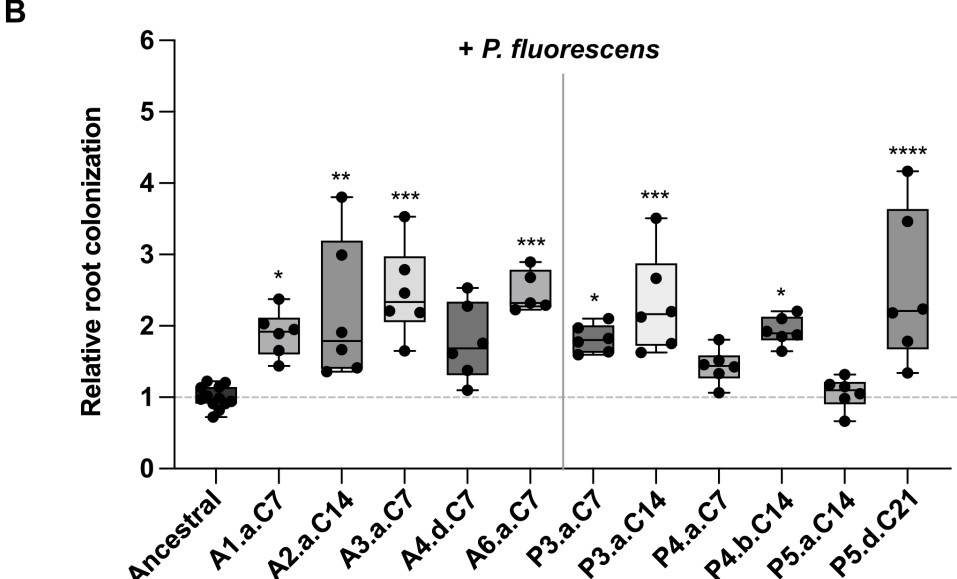

**FIG 3** Tomato root colonization by evolved isolates. Tomato root colonization by evolved isolates from the evolution alone (starts with A) and the evolution with *Pseudomonas* (starts with P) without (A) or with (B) *P. fluorescens* WCS365. Five-to-six-day-old tomato plants were inoculated with ancestral strain or evolved isolates constitutively expressing mKATE2 and incubated for 24 h in a growth chamber. Bacteria were detached and quantified by fluorescence intensity for each plant. Relative root colonization was calculated by dividing the fluorescence intensity of each plant on the average intensity for the ancestral strain of the same experiment. Asterisks indicate statistically significant differences compared to the ancestral strain, $^{*}P < 0.05$; $^{**}P < 0.01$; $^{***}P < 0.001$; $^{****}P < 0.0001$, ANOVA followed by Dunnett's *post hoc* test.

not frequently lead to an increase in colonization of *A. thaliana*, but the increased competitivity in the presence of *P. fluorescens* is conserved on both plants.

### Evolved isolates compete better with other *Pseudomonas* species on roots

As shown in Fig. 1, other *Pseudomonas* species also negatively affect *Bacillus* root colonization. Thus, three of the evolved isolates showing the strongest colonization in the presence of *P. fluorescens* (P3.a.C14, P4.b.C14, and P5.d.C21) were also tested in the

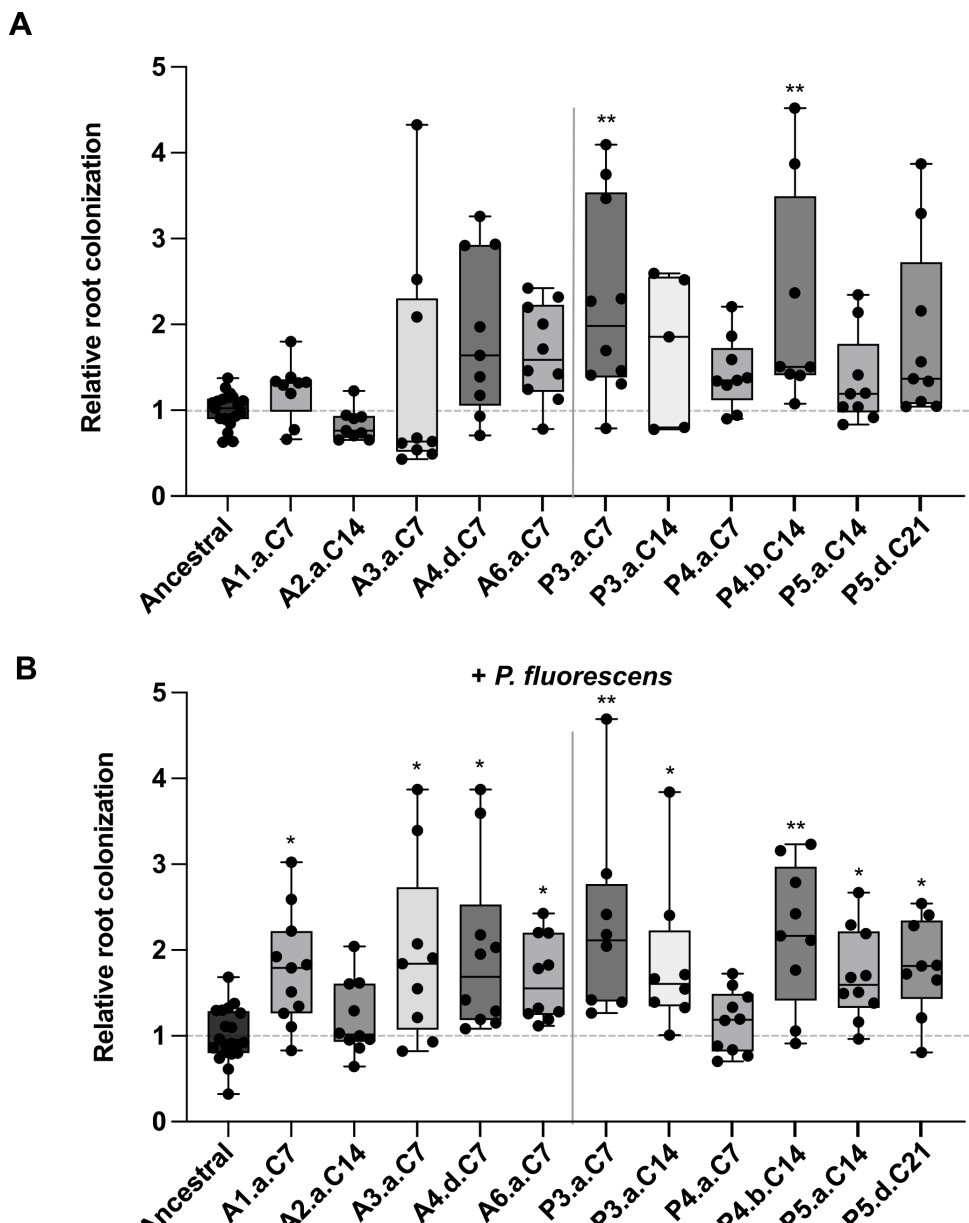

**FIG 4** *A. thaliana* root colonization by evolved isolates *A. thaliana* root colonization by evolved isolates from the BRE (starts with A) and the BRPE (starts with P), without (top panel; A) or with (bottom panel; B) *P. fluorescens* WCS365. Six-day-old plants were inoculated with ancestral strain or evolved isolates constitutively expressing mKATE2 (with or without *P. fluorescens*) and incubated for 24 h in a growth chamber. Whole root pictures were taken using the Zeiss Axio Observer Z1 microscope, the root area was delimited, and relative root colonization was calculated by dividing the mean fluorescence intensity by pixel of each root on the average intensity for the ancestral strain of the same experiment. Asterisks indicate statistically significant differences compared to the ancestral strain, $^*P < 0.05$; $^{**}P < 0.01$, Kruskal–Wallis followed by Dunn's test.

presence of four other *Pseudomonas* strains, inoculated individually at the same time. The species used, *P. protegens* and *Pseudomonas stutzeri,* have with *P. fluorescens* WCS365 a large difference in gene content and thus likely use different antagonistic strategies against *B. subtilis* (46). Despite this, as shown in Fig. 5, we observed that our evolved isolates showed stronger root colonization than the ancestral strain in the presence of either of these *Pseudomonas* strains. This important result hints that the adaptation to *P. fluorescens* led to a general mechanism that also confers advantages in root colonization when another antagonistic member of the same genus is present.

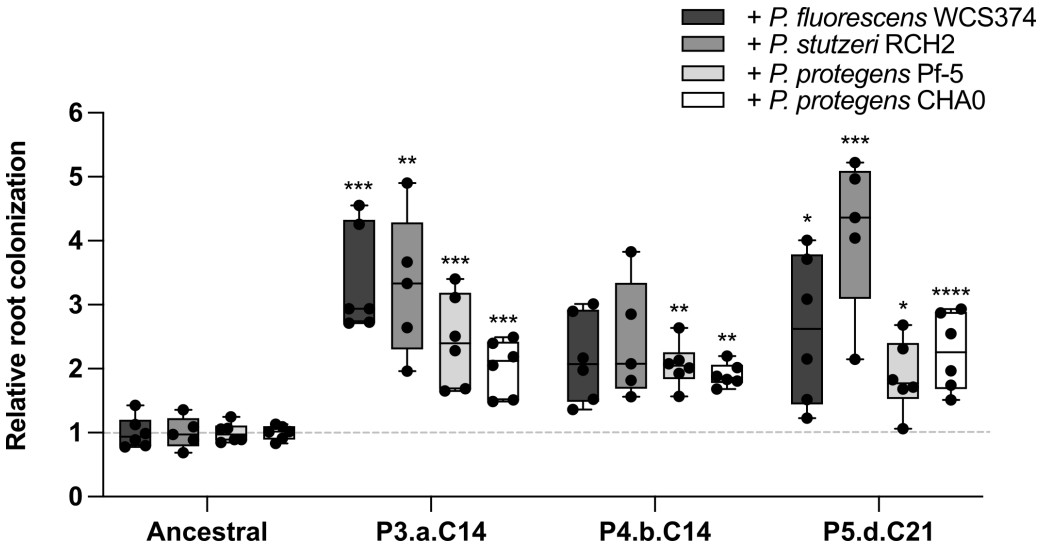

**FIG 5** Mutations in biofilm regulators confer fitness toward various *Pseudomonas* spp. Root colonization in the presence of other *Pseudomonas* spp. Tomato plants were inoculated as previously described, with mKATE2 expressing ancestral strain or evolved isolates and four *Pseudomonas* species. $^{*}P < 0.05$; $^{**}P < 0.01$; $^{***}P < 0.001$; $^{****}P < 0.0001$, ANOVA followed by Dunnett's *post hoc* test (except in 5A for *+P. fluorescens* WCS374 and *+P. stutzeri* RCH2: Kruskal–Wallis followed by Dunn's test).

## Impacts of the non-synonymous mutations on the biofilm regulators functionality

Evolved isolates showing better colonization in the presence of *Pseudomonas* carry non-synonymous mutations in *ywcC, sinR,* and *dxr* (P3.a.C14), *ywcC* and *pksR* (P4.b.C14), and *sinR* and *dxs* (P5.d.C21; see Table 2). For *ywcC*, all the non-synonymous mutations resulted in a frameshift or a stop codon, thus suggesting a loss-of-function, while the non-synonymous mutations present in the other genes were missense variants. Since these non-synonymous mutations provided an advantage when colonizing roots alone and in the presence of different *Pseudomonas*, we examined the deletion mutants for *ywcC, sinR,* and *pksR* and recreated the double mutants as found in the evolved isolates (Fig. 6). Interestingly, only Δ*ywcC* and Δ*pksR* strains showed increased root colonization compared to the wild-type strain, further supporting that the non-synonymous mutations in *ywcC* resulted in a loss-of-function. Conversely, a *sinR* deletion mutant did not confer an increase in root colonization and even negated the advantage provided by Δ*ywcC*, suggesting that the non-synonymous mutations in *sinR* only partially reduced its function. Accordingly, the morphologies of Δ*sinR* strains did not recapitulate the morphologies of the evolved isolates with *sinR* non-synonymous mutations, further supporting this hypothesis (Fig. S1).

## DISCUSSION

In this study, we showed that the antagonism often observed between *Bacillus* spp. and *Pseudomonas* spp. is also observed on plant roots since two different *Pseudomonas* species negatively impacted *Bacillus* root colonization of both *A. thaliana* and tomato (Fig. 1). This result could reflect direct antagonism via antimicrobial mechanisms, like what was previously described using pairwise interaction (21, 22, 36). The weaker inhibitory effect of *P. fluorescens* might be due to its lower number of secondary metabolites (47, 48). However, the antagonism could also be indirect; these *Pseudomonas* could be better adapted to the root microenvironment or capable of establishing faster on the roots, causing a physical displacement of *B. subtilis*.

The directed evolution on tomato roots, performed as a monoculture or in the presence of *P. fluorescens* WCS365, allowed the emergence of different morphotypes of *B. subtilis* through time. Seven out of the total 11 lineages obtained at cycle 21

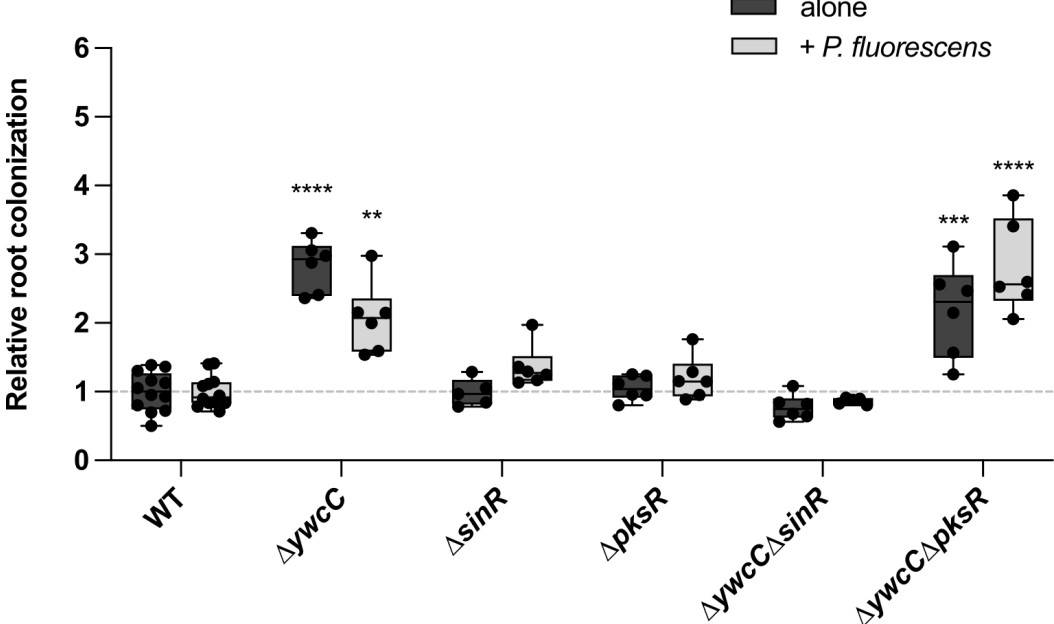

**FIG 6** Deletion of biofilm regulator gene *ywcC* improves tomato roots colonization. Relative colonization of tomato roots by deletion mutants of genes of interest in the study, done as described earlier. Relative root colonization was calculated by dividing the fluorescence intensity of each plant on the average intensity for the ancestral strain of the same experiment. $^{*}P <$ 0.05; $^{**}P < 0.01$; $^{***}P < 0.001$; $^{****}P < 0.0001$, ANOVA followed by Dunnett's *post hoc* test.

had only one morphotype present, while three other lineages had two morphotypes, and one lineage had four morphotypes (Tables S1 and S2). This selection based on morphotype might have induced a bias in which isolates with mutations influencing the biofilm become predominant. However, we also selected many isolates with ancestral-like morphology, and their analysis revealed no mutation, which suggests that we did not miss a significant amount of other types of mutations by selecting using morphotype. Whole-genome sequencing revealed that multiple independent lineages from both evolutions shared non-synonymous mutations in the same few genes, suggesting that evolutionary pressure came from the host and the experimental conditions more than the presence of the competitor. A few of our isolates evolved on tomato also had increased *A. thaliana* root colonization, showing the possibility of one strain being more fit on two different plants. Of note, the presence of 0.05% glycerol in our evolution assay might have relieved the pressure to adapt to specific carbon compounds present in the exudates.

Either *ywcC* or *sinR* were affected by non-synonymous mutations in all independent lineages from both evolution experiments, stressing their importance for root colonization adaptation. Both genes encode for negative regulators of biofilm genes, and their deletion results in hyper-robust biofilm formation (24, 49, 50). The YwcC protein is a TetR-type transcriptional repressor that inhibits *slrA*, a *sinI* paralog. When expressed, SlrA inhibits SinR, thereby derepressing matrix genes (49). Consequently, it was proposed that YwcC responds to a yet unknown environmental signal and alleviates its repression, allowing SlrA to transiently induce matrix production for the whole cell population (49, 51). YwcC also represses poly-y-glutamic acid production, which was shown to play a notable role in plant root colonization (26). Evolved isolates harboring a frameshift mutation in the *ywcC* gene demonstrated enhanced root colonization compared to the ancestral strain, and their morphology suggested robust biofilm formation (Fig. S1). Thus, the mutations acquired during the evolution likely result in a loss of function, which is further supported by the fact that Δ*ywcC* recapitulated *ywcC** evolved isolate phenotypes.

On the contrary, *sinR* loss-of-function did not increase root colonization, even in the presence of *P. fluorescens* or in combination with Δ*ywcC,* pointing toward a milder impact of the non-synonymous mutations on SinR function. Indeed, several *sinR* mutations were previously characterized as displaying a different phenotype than Δ*sinR* and only mildly impacting its function (52, 53). Certain evolved isolates with *sinR* non-synonymous mutation, such as A2.a.C14 from the evolution alone (125C>T; P42L in the DNA binding domain [54]), had no impact on root colonization while displaying a different morphology than the ancestral strain. However, one specific mutation, found in both evolution assays, seemed to enhance root colonization (296T>C; L99S). This mutation was previously shown to impair SinR-L99S binding to DNA and affect its interaction with SinI (52), partially reducing its capacity to repress target genes (55). These observations are in accordance with the observation of similar results of evolved isolates on *A. thaliana* roots in which robust biofilm formers had an increase in root colonization (30).

The BRPE led to the emergence in lineage 4 of a conserved mutation in the *pksR* gene. This gene is part of the *pks* gene cluster, encoding for an enzymatic megacomplex that synthesizes bacillaene, an antimicrobial compound (56). However, this non-synonymous mutation alone (evolved isolate P4.a.C7) did not confer any advantage when colonizing tomato or *A. thaliana* roots, contrary to when this mutation was combined with a non-synonymous mutation *ywcC* (P4.b.C14), suggesting that the latter non-synonymous mutation was responsible for the adaptation.

Finally, non-synonymous mutations in two essential genes involved in the MEP pathway of isoprenoid biosynthesis were identified in two independent isolates of the BRPE showing stronger root colonization (P3.a.C14 and P5.d.C21) (Table 2, *dxr* and *dxs*). While the increased fitness by P3.a.C14 can be attributed to a frameshift mutation in *ywcC*, in P5.d.C21, the *dxs* mutation is accompanied only by a *sinR* non-synonymous mutation that has little to no effect by itself (see P5.a.C14 in Fig. 3B and 4B), indicating this *dxs* non-synonymous mutation had positive effects on the roots. Since *dxs* and *dxr* are essential genes, the mutations are at best a partial loss of function; further experiments will be required to determine the reason behind these beneficial effects.

Overall, directed evolution performed alone (BRE) or in the presence of *P. fluorescens* (BRPE) leads to the parallel evolution of non-synonymous mutations in a few genes that conferred a mild deregulation of biofilm genes. These evolved isolates showed an increased presence on the root compared to the ancestral strain, alone or with various *Pseudomonas* species. Our study indicates that rhizosphere fitness in the presence of a competitor is closely linked to robust root colonization and physical establishment.

## MATERIALS AND METHODS

### Bacterial strains and culture conditions

The strains used in this study are listed in Table S3. *B. subtilis* strains were routinely cultivated in lysogeny broth (Luria-Bertani [LB]; 1% [wt/vol] tryptone, 0.5% [wt/vol] yeast extract, 0.5% [wt/vol] NaCl) at 37°C in agitation for 3 h. *Pseudomonas* strains were cultivated in LB at 30°C in agitation for 3 h. Experimental evolution and all root colonization assays were performed in a half-strength Murashige and Skoog (MS) basal medium (2.22 g L$^{-1}$, Sigma) supplemented with 0.05% glycerol. When necessary, antibiotics were used at the following concentrations: spectinomycin (100 µg·mL$^{-1}$), kanamycin (10 µg·mL$^{-1}$), chloramphenicol (5 µg·mL$^{-1}$), and erythromycin (1 µg·mL$^{-1}$).

### Seedling preparation

*A. thaliana* ecotype Col-0 and tomato (*Solanum lycopersicum*, Rutgers, McKenzie seeds, California, and Greta's Family Gardens, California) were used in this study. Seeds were surface sterilized with 70% [vol/vol] ethanol followed by 0.3% [vol/vol] sodium hypochlorite then rinsed three times with sterile Milli-Q water and plated on a Murashige and Skoog basal medium (4.44 g L$^{-1}$, Sigma) 0.8% [wt/vol] agar supplemented with 0.05% [wt/vol] glucose. Plates were incubated in a growth chamber with a day/night

cycle of 12 h light at 25°C and 12 h dark at 20°C, *A. thaliana* seedlings for 7 days and tomato seedlings for 5 or 6 days.

## Strain constructions

All genetically modified strains were made by transferring genetic constructions into NCIB 3610 or evolved isolates using SPP1-mediated generalized transduction (57). All deletion mutants used in this study were purchased from the Bacillus Genetic Stock Center collection (http://www.bgsc.org) in the *B. subtilis* 168 background. *Pseudomonas* strains *P. fluorescens* WCS374, *P. stutzeri* RCH2, and *P. protegens* CHA0 were obtained from Cara Haney (UBC).

## Experimental evolution

Tomato seedlings were transferred to six-well plates containing 4 mL of 1/2 MS + 0.05% [vol/vol] glycerol (two floating seedlings/plate in opposite corners to prevent contamination between lineages). *B. subtilis* NCIB 3610 culture was washed in phosphate-buffered saline (PBS) and inoculated at a final optical density at 600 nm ($OD_{600}$) of 0.02. Plates were put on a shaker at 90 rpm in the growth chamber in the same conditions as for the seedling preparation. For evolution alone (BRE), after 24 h, the root was detached from the stem using tweezers, rinsed in PBS, and sonicated at 30% amplitude for 30 s in a microtube containing 1 mL PBS. One hundred thirty-three microliters of the bacterial suspension was inoculated on a new plant, and the suspension was diluted and plated to follow the number of bacteria transferred every cycle. For evolution with *Pseudomonas* (BRPE), after 24 h of *B. subtilis* root colonization, *P. fluorescens* WCS365 was inoculated in each well at a final $OD_{600}$ of 0.0002, which correspond approximately to a 100:1 *Bacillus*:*Pseudomonas* ratio, and reincubated in the growth chamber for 24 h. Any lower *Bacillus*:*Pseudomonas* ratio led to the rapid disappearance of the *Bacillus* population, suggesting an evolutionary pressure (not shown). The root was then detached, washed, and sonicated (40% amplitude, 60 s), which killed 99% of the *Pseudomonas*. One hundred thirty-three microliters of the bacterial suspension was then inoculated on a new plant, and the suspension was diluted and plated to follow the number of bacteria transferred every cycle. For cycles 3–21, *P. fluorescens* was added after 24 h at each cycle at a final $OD_{600}$ of 0.00002 or approximately $7.3 \times 10^3$ CFU/mL ($OD_{600}$ of 0.6 = $2.2 \times 10^3$ CFU/mL). Considering the *B. subtilis* CFU number on the root (see Fig. 2C), this *P. fluorescens* inoculum corresponds to a *Bacillus*:*Pseudomonas* ratio of around 1:10 to 100:1, depending on the lineage and the cycle.

## Whole-genome sequencing

Genomic DNA was extracted from 9 mL of 3 h cultures using the Monarch Genomic DNA Purification Kit. Paired-end (evolution alone) or single-end (evolution with *Pseudomonas*) libraries were prepared using the NEBNext Ultra II DNA Library Prep with Sample Purification Beads. Fragment reads were generated on an Illumina NextSeq sequencer using TG NextSeq 500/550 High Output Kit v2 (300 cycles) (RNOmique Platform, UdeS). The methodology for processing the reads drew inspiration from the GenPipes framework (58). We first implemented a quality-based trim of the reads using fastp 0.21.0 (59), setting the parameters to --cut_right --cut_window_size 4 --cut_mean_quality 30 --length_required 30. The trimmed reads were subsequently mapped to the reference *Bacillus subtilis* 3610 and its plasmid pBS32, procured from GenBank (CP020102.1 and CP020103.1), using BWA-MEM v0.7.10 (60) with no modification of the default parameters. Upon alignment, we sorted the reads with Picard v1.123, available online at https://broadinstitute.github.io/picard/, and realignment was executed with the GATK v3.7 RealignerTargetCreator and IndelRealignerSingle tools (61). Following the identification and marking of duplicate reads with Picard v1.123, we utilized GATK v3.7 for haplotype identification. Subsequently, their impacts were evaluated using SnpEff v4.1 (62).

## Colonization assays

Evolved isolates formed a very robust biofilm on the root surface, so CFU counting to quantify root colonization was unreliable since clumps of bacteria could not be separated by sonication. We developed two methods that allowed us to quantify relative root colonization regardless of clumps, using a constitutive fluorescent reporter that was transduced into all strains tested. For tomato, *B. subtilis* and evolved mutants were transduced with the *amyE*::P$_{hyperspank}$-*mkate2* and then inoculated on roots as described in the evolution assay on tomato plants. For the condition in the presence of *Pseudomonas*, both species were inoculated at the same time and the same OD (OD$_{600}$ = 0.02). After 24 h, the roots were detached and transferred in a microtube containing 500 µL PBS with three 3-mm glass beads. Roots were vortexed for 30 s and sonicated at 40% amplitude for 20 s. Two hundred microliters of the supernatant was transferred to a 96-well plate, and the fluorescent intensity was measured with a TECAN Spark monochromator with an excitation wavelength of 590 nm and emission wavelength of 638 nm, bandwidth 20. A calibration curve using this method is shown in Fig. S2. For *A. thaliana*, the assays were performed in 48-well plates containing 280–290 µL of media. Bacteria were either detached from the root using sonication (amplitude 30%, 10 pulses, 1 s pulse and 1 s rest time), diluted, and plated on solid LB media to allow CFU counting, or bacterial colonization was quantified in microscopy (see Microscopy).

## Microscopy

To visualize bacteria on the root surface for Fig. 1, seedlings were examined with a Zeiss Axio Observer Z1 microscope equipped with a 20×/0.8 Plan-Apochromat objective, and whole root pictures were taken with a Zeiss Axiocam 506 mono. The fluorescence signal was detected using a Cy3 filter (ex: 545/25, em: 605/70). All images were taken at the same exposure time, processed identically, and prepared for presentation using ImageJ. Each image is representative of at least nine root colonization assays performed in three independent experiments. Quantification was done using Fiji. The root area was delimited, and relative root colonization was calculated by dividing the mean fluorescence intensity on the average intensity for the ancestral strain of the same experiment. Fig. S1 was obtained using the stereo microscope Leica M165 FC with the Leica MC170 HD camera.

## Statistical analysis

Statistical analyses were performed using GraphPad Prism 9. Comparisons were done as specified in the figure legends. Normality was tested using the Shapiro-Wilk test.

### ACKNOWLEDGMENTS

We thank the members of the Beauregard laboratory and Rodrigue laboratory for their helpful discussions. We also thank Daniel Garneau for the technical advice and processing of images. We thank Julie Beaudin for the critical reading of the manuscript and Cara Haney, Roberto Kolter, and Richard Losick for the kind gift of strains.

This work was supported by NSERC discovery grant RGPIN-2020-07057 to P.B.B.

### AUTHOR AFFILIATIONS

[1]Département de biologie, Faculté des sciences, Université de Sherbrooke, Sherbrooke, Québec, Canada
[2]Département de Génie Biologique, Université de Technologie de Compiègne, Compiègne, France

### AUTHOR ORCIDs

Maude Pomerleau http://orcid.org/0009-0009-9885-1615

Sébastien Rodrigue  http://orcid.org/0000-0002-5366-7234

Pascale B. Beauregard  http://orcid.org/0000-0003-2947-0500

## FUNDING

| Funder | Grant(s) | Author(s) |
| --- | --- | --- |
| Natural Sciences and Engineering Research Council of Canada (NSERC) | RGPIN-2020-07057 | Pascale B. Beauregard |

## AUTHOR CONTRIBUTIONS

Maude Pomerleau, Conceptualization, Investigation, Methodology, Writing – original draft | Vincent Charron-Lamoureux, Conceptualization, Investigation, Writing – review and editing | Lucille Léonard, Investigation | Frédéric Grenier, Formal analysis, Writing – review and editing | Sébastien Rodrigue, Supervision, Writing – review and editing | Pascale B. Beauregard, Conceptualization, Funding acquisition, Supervision, Writing – original draft, Writing – review and editing

## DATA AVAILABILITY

The sequencing data files of evolved isolates were submitted to the NCBI Sequence Read Archive (SRA) database under Bioproject ID PRJNA1028130.

## ADDITIONAL FILES

The following material is available online.

### Supplemental Material

**Supplemental material (mSystems00843-23-s0001.pdf).** Supplemental figures and tables.

### Open Peer Review

**PEER REVIEW HISTORY (review-history.pdf).** An accounting of the reviewer comments and feedback.

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
