## [Reviewer comments · mSystems]

Adaptive Laboratory Evolution reveals regulators involved in repressing biofilm development as key players in *B. subtilis* root colonization

Maude Pomerleau, Vincent Charron-Lamoureux, Lucille Léonard, Frédéric Grenier, Sébastien Rodrigue, and Pascale Beauregard

Corresponding Author(s): Pascale Beauregard, Université de Sherbrooke

Review Timeline:

Submission Date:	August 10, 2023
Editorial Decision:	September 14, 2023
Revision Received:	October 31, 2023
Editorial Decision:	November 29, 2023
Revision Received:	November 30, 2023
Accepted:	December 4, 2023

Editor: Aurélie Deveau

Reviewer(s): Disclosure of reviewer identity is with reference to reviewer comments included in decision letter(s). The following individuals involved in review of your submission have agreed to reveal their identity: Ákos T. Kovács (Reviewer #1); Yunrong Chai (Reviewer #2)

Transaction Report:

DOI: <https://doi.org/10.1128/msystems.00843-23>

September 14, 2023

Mrs. Maude Pomerleau
Universite de Sherbrooke
Departement de Biologie
2500 boul. Université
Sherbrooke, QC J1K 2R1
Canada

Re: mSystems00843-23 (Adaptive Laboratory Evolution reveals biofilm regulating genes as key players in *B. subtilis* root colonization)

Dear Mrs. Maude Pomerleau:

Thank you for submitting your manuscript to mSystems. We have completed our review and I am pleased to inform you that, in principle, we expect to accept it for publication in mSystems. However, acceptance will not be final until you have adequately addressed the reviewer comments.

Preparing Revision Guidelines

Please return the manuscript within 60 days; if you cannot complete the modification within this time period, please contact me. If you do not wish to modify the manuscript and prefer to submit it to another journal, please notify me of your decision immediately so that the manuscript may be formally withdrawn from consideration by mSystems.

Sincerely,

Aurélie Deveau

Editor, mSystems

Journals Department
Reviewer comments:

Reviewer #1 (Comments for the Author):

Maude Pomerleau and colleagues present a compelling story on how increased biofilm matrix production benefit *Bacillus subtilis* to colonize plant root in the presence or absence of competing *Pseudomonas*. Specifically, they perform an adaptive laboratory evolution using a strain of *B. subtilis* on tomato roots, and after repeated rounds of artificial dispersion (sonication) and reinoculation, they observe that adapted isolated have similar properties, these colonize the plant roots at higher abundance, and also able to establish stronger when *Pseudomonas* is subsequently inoculated. The experiments are clearly described (with minor details required depicted below), the manuscript is clearly written (with only minor comments), thus all in all, this is nice contribution to the field working on *Bacillus*-plant interaction. The experimental setup is novel; these insights confirm previous publications, and it provide a systemic insight into the observed mutations. Even if there is no strong difference in the evolution of *B. subtilis* when *Pseudomonas* is added subsequently after establishment at each round of root colonization, there are interesting insights on how increased biofilm formation might be a powerful method to enhance competitiveness of *B. subtilis* against inhibiting bacterial strains, like *Pseudomonas protegens*.

I have only minor comments that can be all easily addressed in the manuscript text without additional experimental approach. Great work!

Major:

SNP refers to single nucleotide polymorphism, a genomic variant at a single base position in the DNA. This includes nucleotide alteration in non-coding genes, and both synonymous and non-synonymous (amino acid changing) mutations. The authors use SNP in this manuscript for non-synonymous mutations, which is incorrect. Please use the term non-synonymous mutation term where it is applicable. Please be careful of this in the abstract and from line 160 and forward.

Results in lines 113-115 should be discussed by comparing to in vitro biofilm studies done between *Bacillus subtilis* and diverse *Pseudomonas*, both specific examples and generally: <https://doi.org/10.1101/2023.07.18.549276>

The experimental setup should be briefly mentioned in the appropriate section of the results, as it comes way too late in the materials and methods. I do not recommend to explain all details, but general setup. However (see below), the major difference between BRE and BRPE should be clearly mentioned, namely the difference in time and amplitude of sonication used. This might alter the selection strength.

Line 166: Can you report the estimated difference or potential difference in population size? That might have huge impact, as previously reported

Line 173: rephrase "took longer". I am unsure what this refers to. Do you refer to that there is a different selection pressure? Smaller population size, milder selection?

Line 179: Do you observe a correlation of these mutations with certain morphotypes?

Line 264-266 should be discussed later in the discussion; see Richter et al, SinR modifications display a not entirely loss of function mutations, e.g. motility is required which is otherwise fully reduced in sinR mutant. This should be mentioned.

Line 317: luckily, the exact same amino acid change in SinR, L99S has been previously found as suppressor mutation during biofilm development of strains lacking motility, see Richter et al: <https://doi.org/10.1186/s12862-018-1266-2>
Also, that publication provides detailed characterization of the given mutation (see Fig 3 in the paper)

Line 372: Please indicate if only root was submerged or the whole plant, or plant was floating on top of the medium

Line 380: When were *Pseudomonas* inoculated, which stage of *B.subtilis* colonization? Few hours or days?

Major concern that should be mentioned in the results and discussion also that the different sonication times and amplitudes were used during the two types of experiment (BRPE and BRE), which might also influence the different evolution (higher

mutations) in BRPE?

Fig 1: Note, Bacillus is peritrichous, use the appropriate symbol (search for Salmonella in BioRender). Please use different colors for Bacillus and Pseudomonas, as it is hard to differentiate between the symbols, if they are so small.

The raw sequencing data files must be uploaded to SRA to fulfill the open data availability criteria.

Minor:

Line 1: The title should be adjusted. Please, do not use "biofilm regulating genes". Genes are not regulating, but the protein products. Better to use: Adaptive Laboratory Evolution reveals mutations in genes encoding regulators of biofilm development during Bacillus subtilis root colonization.

Line 24: replace the term "negative biofilm regulator gene", as there is no "negative biofilm", a gene cannot regulate, but the protein product, also a regulator cannot regulate negatively. Use: "In all independent lineages, whole genome resequencing revealed non-synonymous mutations in genes ywC or sinR encoding regulators involved in repressing biofilm development."

Line 32: remove "even"

Line 33: what was "monitored"? I do not see in the material method any parameters being monitored. Do you mean defined environment?

Line 34: correct to "were more effective at colonizing plant roots than the ancestor strain."

Line 35: Sequencing itself will not tell this just provide potential list or motivate such, please reword the start of the sentence

Line 39: do not use cognitive terms for bacteria and reword "clever strategies employed by Bacillus subtilis". Rather: "Our research sheds light on the mutational paths selected in Bacillus subtilis to thrive"

Line 63: reword, it is a very difficult sentence to understand, especially "exposed or not to other microorganisms"

Line 77: correct to "Adaptive Laboratory Evolution (ALE) experiment is a powerful tool"

Line 96: correct to "B. subtilis was either challenged with Pseudomonas or not."

Line 98: do not use negative regulators, also genes are not regulators, but the encoded proteins (see above)

Line 131: Please indicate "on tomato roots before being actively dispersed by sonication and diluted samples were transferred to a new root every". This is an important experimental difference compared to previous ALE studies using Bacillus and plant roots.

Line 138: please do not use "data not shown". Please add graph or microscopy data on this observation in the supplementary

Line 142: By Bacillus or both? Indicated whether the uptrend is due to both organisms

Paragraph starting in line 150: Clearly mention that only B.subtilis was assayed here

Line 154: adjust to "The morphologies of colonies seeded from cultures from the frozen stocks were then"

Line 158: I am unsure what "clones of at least one other" refers to. Do you refer to within lineage or between lineages matching (which suggests that mutations existed in starting cultures or cross contamination).

Line 211: "adapts differently" is not fully recapitulate those observations, rather, indicate that different mutations were observed, the different adaptation was not demonstrated in that paper

Line 237: you mention 4 other Pseudomonas species, but I see only 3 here

Line 247, 270: be more precise what "as previously described" refers to. Is it a previous paper, or previous experiment above?

Supplementary:

"Lab stock" labels should be replaced with detailed information where these originate from or how they were constructed (e.g. amyE:hyperspank-mKATE2). Laboratory stock provides insufficient information on these strains.

Reviewer #2 (Comments for the Author):

In this study by Pomerleau et al, the authors designed laboratory evolution assays to understand how the rhizobacterium Bacillus subtilis adapt to the roots of Arabidopsis and tomato, and compete with other rhizobacteria on the roots, specifically Pseudomonas species by genetically altering related genes. These direct evolution experiments are powerful and creative in understanding bacterial competition, antagonism, and adaptation at the genetic level using whole genome sequencing. In previous studies, similar evolution assays have been used to understand what types of genetic adaptations allow B. subtilis to colonize the roots better. The experiments in this study were reasonably designed and some of the findings are interesting. One of the important findings from this study seems to be that biofilm formation plays an important role in root colonization and bacterial competition on roots since multiple SNPs were identified in the two known biofilm regulator genes, ywC and sinR, consistent with findings from other published studies but by using novel evolution assays. This study represents a more systematic and unbiased fashion, leading to similar conclusions. I have two suggestions, especially how some of sequencing results can be better organized and presented.

1. Identification of SNPs in the evolved B. subtilis isolates are important results to understand the environmental selection occurred to, and adaptation by, B. subtilis, the current way of presenting the SNP results seems less informative, especially involving different lineages and when those SNPs appear and how they are inherited. I wonder if a comprehensive table or heatmap or flowchart may work better.
2. It seems to me that the selection of evolved isolates for genome sequencing has a precondition/bias: many sequenced isolate

show alteration of morphologies. This could lead to bias in identifying genetic mutations occurred in evolution assays, e.g. SNPs in biofilm regulator genes become predominant. The authors should acknowledge and discuss about it.

3. In figure 2B-C, it is quite interesting to see that the cfu of *Bacillus subtilis* is trending up significantly comparing early and late cycles. I wonder what happens to the relative ratio of *Bacillus* vs *Pseudomonas*. Does *B. subtilis* become dominant after 20 plus cycles ?

Line 50, ...as a plant growth-promoting rhizobacterium.

Lines 262-266, 312- 321, the differences between deletion mutations in *sinR* and certain SNPs in *sinR* and their differential impacts were also investigated in a previous study (Chai et al. 2010, *Genes Dev.* 24:754-765.), which may provide some explanation toward the authors' observations in this study.

Reviewer #1 (Comments for the Author):

Maude Pomerleau and colleagues present a compelling story on how increased biofilm matrix production benefit *Bacillus subtilis* to colonize plant root in the presence or absence of competing *Pseudomonas*. Specifically, they perform an adaptive laboratory evolution using a strain of *B. subtilis* on tomato roots, and after repeated rounds of artificial dispersion (sonication) and reinoculation, they observe that adapted isolated have similar properties, these colonize the plant roots at higher abundance, and also able to establish stronger when *Pseudomonas* is subsequently inoculated. The experiments are clearly described (with minor details required depicted below), the manuscript is clearly written (with only minor comments), thus all in all, this is nice contribution to the field working on *Bacillus*-plant interaction. The experimental setup is novel; these insights confirm previous publications, and it provide a systemic insight into the observed mutations. Even if there is no strong difference in the evolution of *B. subtilis* when *Pseudomonas* is added subsequently after establishment at each round of root colonization, there are interesting insights on how increased biofilm formation might be a powerful method to enhance competitiveness of *B. subtilis* against inhibiting bacterial strains, like *Pseudomonas protegens*.

I have only minor comments that can be all easily addressed in the manuscript text without additional experimental approach. Great work! **Thanks!**

Major:

SNP refers to single nucleotide polymorphism, a genomic variant at a single base position in the DNA. This includes nucleotide alteration in non-coding genes, and both synonymous and non-synonymous (amino acid changing) mutations. The authors use SNP in this manuscript for non-synonymous mutations, which is incorrect. Please use the term non-synonymous mutation term where it is applicable. Please be careful of this in the abstract and from line 160 and forward.

This was corrected.

Results in lines 113-115 should be discussed by comparing to in vitro biofilm studies done between *Bacillus subtilis* and diverse *Pseudomonas*, both specific examples and generally: <https://doi.org/10.1101/2023.07.18.549276>

This was added, it brings an interesting point (lines 118-120).

The experimental setup should be briefly mentioned in the appropriate section of the results, as it comes way too late in the materials and methods. I do not recommend to explain all details, but general setup. However (see below), the major difference between BRE and BRPE should be clearly mentioned, namely the difference in time and amplitude of sonication used. This might alter the selection strength.

We added more details in the result section (line 135 and lines 138-141).

Line 166: Can you report the estimated difference or potential difference in population size? That might have huge impact, as previously reported

Population size in the two evolution is in direct proportion with the CFU on the root (Fig 2BC), since the inoculum for each cycle consist in 13,3% of the cells present on the root at the end of the previous cycle. Thus, in some cycles BRPE population size is significantly smaller than BRE population size, but not in most. We added a sentence specifying this (lines 174-175)

Line 173: rephrase "took longer". I am unsure what this refers to. Do you refer to that there is a different selection pressure? Smaller population size, milder selection?

This sentence was reformulated for more precision (lines 181-183).

Line 179: Do you observe a correlation of these mutations with certain morphotypes?

We observe a correlation with a "rough" morphotype and mutations in *ywcC*, but we could not correlate mutations in *sinR* with a single morphotype. Precisions were added (line 188-190).

Line 264-266 should be discussed later in the discussion; see Richter et al, *SinR* modifications display a not entirely loss of function mutations, e.g. motility is required which is otherwise fully reduced in *sinR* mutant. This should be mentioned.

This was added in the discussion (line 334-336).

Line 317: luckily, the exact same amino acid change in *SinR*, L99S has been previously found as suppressor mutation during biofilm development of strains lacking motility, see Richter et

al: <https://doi.org/10.1186/s12862-018-1266-2>

Also, that publication provides detailed characterization of the given mutation (see Fig 3 in the paper)
Information about the effect of this mutation was added (lines 340-341).

Line 372: Please indicate if only root was submerged or the whole plant, or plant was floating on top of the medium

This detail was added (line 398).

Line 380: When were *Pseudomonas* inoculated, which stage of *B. subtilis* colonization? Few hours or days?
Pseudomonas is added 24h hours after *B. subtilis* is added on the root, so during biofilm formation. The sentence was re-written for more clarity (line 405).

Major concern that should be mentioned in the results and discussion also that the different sonication times and amplitudes were used during the two types of experiment (BRPE and BRE), which might also influence the different evolution (higher mutations) in BRPE?

This precision was added in the result, since it flowed better in the text there (lines 170-172).

Fig 1: Note, *Bacillus* is peritrichous, use the appropriate symbol (search for *Salmonella* in BioRender). Please use different colors for *Bacillus* and *Pseudomonas*, as it is hard to differentiate between the symbols, if they are so small.

Fig 2 has been modified accordingly.

The raw sequencing data files must be uploaded to SRA to fulfill the open data availability criteria.

The sequencing data files were submitted to Bioproject (BioProject ID PRJNA1028130) and they will be available upon publication (line 435).

Minor:

Line 1: The title should be adjusted. Please, do not use "biofilm regulating genes". Genes are not regulating, but the protein products. Better to use: Adaptive Laboratory Evolution reveals mutations in genes encoding regulators of biofilm development during *Bacillus subtilis* root colonization.

The title was modified for "Adaptive Laboratory Evolution reveals regulators involved in repressing biofilm development as key players in *B. subtilis* root colonization »

Line 24: replace the term "negative biofilm regulator gene", as there is no "negative biofilm", a gene cannot regulate, but the protein product, also a regulator cannot regulate negatively. Use: "In all independent lineages, whole genome resequencing revealed non-synonymous mutations in genes *ywcC* or *sinR* encoding regulators involved in repressing biofilm development."

This sentence was modified accordingly (lines 24-26).

Line 32: remove "even"

This was removed (line 33).

Line 33: what was "monitored"? I do not see in the material method any parameters being monitored. Do you mean defined environment?

Yes indeed, this was corrected (line 34).

Line 34: correct to "were more effective at colonizing plant roots than the ancestor strain."

This was corrected (line 35-36).

Line 35: Sequencing itself will not tell this just provide potential list or motivate such, please reword the start of the sentence

This was corrected (line 36).

Line 39: do not use cognitive terms for bacteria and reword "clever strategies employed by *Bacillus subtilis*". Rather: "Our research sheds light on the mutational paths selected in *Bacillus subtilis* to thrive"

This was corrected (line 40).

Line 63: reword, it is a very difficult sentence to understand, especially "exposed or not to other microorganisms"

This was indeed bad English, and was corrected (line 63-64).

Line 77: correct to "Adaptive Laboratory Evolution (ALE) experiment is a powerful tool"

We prefer to use the plural of experiments, so we modified the sentence in " Adaptive Laboratory Evolution (ALE) experiments are a powerful tool for studying microbial adaptation to specific environments."

Line 96: correct to "*B. subtilis* was either challenged with *Pseudomonas* or not."

This was corrected (line 98).

Line 98: do not use negative regulators, also genes are not regulators, but the encoded proteins (see above)

This was corrected (lines 99 -101).

Line 131: Please indicate "on tomato roots before being actively dispersed by sonication and diluted

samples were transferred to a new root every". This is an important experimental difference compared to previous ALE studies using *Bacillus* and plant roots.

This was corrected (line 135).

Line 138: please do not use "data not shown". Please add graph or microscopy data on this observation in the supplementary

No data were kept on the evolution attempts with *P. protegens* except "*B. subtilis* was outcompeted", which is our mistake. We deleted all mention of these assays to avoid having to use "data not shown" (line 144).

Line 142: By *Bacillus* or both? Indicated whether the uptrend is due to both organisms

This was specified (line 147).

Paragraph starting in line 150: Clearly mention that only *B. subtilis* was assayed here

This was specified (line 156).

Line 154: adjust to "The morphologies of colonies seeded from cultures from the frozen stocks were then"

This was corrected (lines 160-161).

Line 158: I am unsure what "clones of at least one other" refers to. Do you refer to within lineage or between lineages matching (which suggests that mutations existed in starting cultures or cross contamination).

It is indeed within lineage, this was specified (line 164).

Line 211: "adapts differently" is not fully recapitulate those observations, rather, indicate that different mutations were observed, the different adaptation was not demonstrated in that paper

This was corrected (line 227-228).

Line 237: you mention 4 other *Pseudomonas* species, but I see only 3 here

Indeed, there are 4 strains from 3 species since there are 2 strains of *P. protegens*; this was specified (line 254).

Line 247, 270: be more precise what "as previously described" refers to. Is it a previous paper, or previous experiment above?

Previous experiment – this was modified (line 264 and 288).

Supplementary:

"Lab stock" labels should be replaced with detailed information where these originate from or how they were constructed (e.g. amyE:hyperspank-mKATE2). Laboratory stock provides insufficient information on these strains.

Additional precision were provided in the strain table.

Reviewer #2 (Comments for the Author):

In this study by Pomerleau et al, the authors designed laboratory evolution assays to understand how the rhizobacterium *Bacillus subtilis* adapt to the roots of *Arabidopsis* and tomato, and compete with other rhizobacteria on the roots, specifically *Pseudomonas* species by genetically altering related genes. These direct evolution experiments are powerful and creative in understanding bacterial competition, antagonism, and adaptation at the genetic level using whole genome sequencing. In previous studies, similar evolution assays have been used to understand what types of genetic adaptations allow *B. subtilis* to colonize the roots better. The experiments in this study were reasonably designed and some of the findings are interesting. One of the important findings from this study seems to be that biofilm formation plays an important role in root colonization and bacterial competition on roots since multiple SNPs were identified in the two known biofilm regulator genes, *ywcC* and *sinR*, consistent with findings from other published studies but by using novel evolution assays. This study represents a more systematic and unbiased fashion, leading to similar conclusions. I have two suggestions, especially how some of sequencing results can be better organized and presented.

1. Identification of SNPs in the evolved *B. subtilis* isolates are important results to understand the environmental selection occurred to, and adaptation by, *B. subtilis*, the current way of presenting the SNP results seems less informative, especially involving different lineages and when those SNPs appear and how they are inherited. I wonder if a comprehensive table or heatmap or flowchart may work better.

We added lines to Table 1 and 2 to specify what lineage and at which cycle the evolved strains were isolated. We also used scales of gray to clarify at which cycles the mutations had emerged in the table describing all the mutants (Table S1 and S2). We think the summary and comprehensive tables are now fairly clear, but if the reviewer has any additional suggestion, we will gladly incorporate them.

2. It seems to me that the selection of evolved isolates for genome sequencing has a precondition/bias: many sequenced isolate show alteration of morphologies. This could lead to bias in identifying genetic mutations occurred in evolution assays, e.g. SNPs in biofilm regulator genes become predominant. The authors should acknowledge and discuss about it.

Discussion about this bias was added (line 307-310).

3. In figure 2B-C, it is quite interesting to see that the cfu of *Bacillus subtilis* is trending up significantly

comparing early and late cycles. I wonder what happens to the relative ratio of Bacillus vs Pseudomonas. Does B. subtilis become dominant after 20 plus cycles ?

Since we add a fixed number of Pseudomonas at each cycle, B. subtilis is always dominant, only the ratio changes. We explain it in details in the Method section (lines 405-416)

Line 50, ...as a plant growth-promoting rhizobacterium.

This was corrected (line 51).

Lines 262-266, 312- 321, the differences between deletion mutations in sinR and certain SNPs in sinR and their differential impacts were also investigated in a previous study (Chai et al. 2010, Genes Dev. 24:754-765.), which may provide some explanation toward the authors' observations in this study.

Indeed, this was added in the discussion (lines 332-343).

Re: mSystems00843-23R1 (Adaptive Laboratory Evolution reveals regulators involved in repressing biofilm development as key players in *B. subtilis* root colonization)

Dear Mrs. Maude Pomerleau:

Thank you for the privilege of reviewing your work. We are happy to accept your manuscript if you make the minor modifications given below.

Please correct the following points: first, to comply with ASM policy, please include in a specific paragraph entitled "Data availability" at the end of the Material and Method section all the information regarding data availability, repository... These information were included in your manuscript but should be moved to this section. Second, regarding fig2A, please correct as suggested by reviewer: it looks odd to see flagellated cells depicted on the roots in Fig 2A, while specifically, the excellent mBio paper (<https://doi.org/10.1128/mbio.01664-16>) from the same lab demonstrated the switch to non-motile lifestyle during plant colonization. The colors are now distinctive, I suggest indicating cells without flagella when attached to the root.

Revision Guidelines

Sincerely,
Aurélié Deveau
Editor
mSystems

Reviewer #1 (Comments for the Author):

Thank you for carefully addressing all comments. Only a minor remaining comment: it looks odd to see flagellated cells depicted on the roots in Fig 2A, while specifically, the excellent mBio paper (<https://doi.org/10.1128/mbio.01664-16>) from the same lab demonstrated the switch to non-motile lifestyle during plant colonization. The colors are now distinctive, I suggest indicating cells without flagella when attached to the root.

Reviewer #2 (Comments for the Author):

This work using laboratory evolution to study the interactions between *Bacillus subtilis* and *Pseudomonas* species, two well-known PGPR bacteria, is quite interesting and novel. In this revised manuscript, the authors have successfully addressed comments and issues i raised in the previous review. I am satisfied with the revisions. The manuscript is also well written. I have no further comments.

Please correct the following points : first, to comply with ASM policy, please include in a specific paragraph entitled "Data availability" at the end of the Material and Method section all the information regarding data availability, repository... These information were included in your manuscript but should be moved to this section.
Data availability paragraph has been added. Lines 418-420.

Reviewer #1 (Comments for the Author):

Thank you for carefully addressing all comments. Only a minor remaining comment: it looks odd to see flagellated cells depicted on the roots in Fig 2A, while specifically, the excellent mBio paper (<https://doi.org/10.1128/mbio.01664-16>) from the same lab demonstrated the switch to non-motile lifestyle during plant colonization. The colors are now distinctive, I suggest indicating cells without flagella when attached to the root.
The figure 2A has been adjusted accordingly. Cells attached to the root have been changed.

Reviewer #2 (Comments for the Author):

This work using laboratory evolution to study the interactions between *Bacillus subtilis* and *Pseudomonas* species, two well-known PGPR bacteria, is quite interesting and novel. In this revised manuscript, the authors have successfully addressed comments and issues i raised in the previous review. I am satisfied with the revisions. The manuscript is also well written. I have no further comments.
Thank you!

Re: mSystems00843-23R2 (Adaptive Laboratory Evolution reveals regulators involved in repressing biofilm development as key players in *B. subtilis* root colonization)

Dear Mrs. Maude Pomerleau:

Your manuscript has been accepted, and I am forwarding it to the ASM production staff for publication. Your paper will first be checked to make sure all elements meet the technical requirements. ASM staff will contact you if anything needs to be revised before copyediting and production can begin. Otherwise, you will be notified when your proofs are ready to be viewed.

Featured Image Submissions: If you would like to submit a potential Featured Image, please email a file and a short legend to mSystems@asmusa.org. Please note that we can only consider images that (i) the authors created or own and (ii) have not been previously published. By submitting, you agree that the image can be used under the same terms as the published article. File requirements: square dimensions (4" x 4"), 300 dpi resolution, RGB colorspace, TIF file format.

Sincerely,
Aurélie Deveau
Editor
mSystems